# The Australian Feeding Infants and Toddler Study (OzFITS 2021): Breastfeeding and Early Feeding Practices

**DOI:** 10.3390/nu14010206

**Published:** 2022-01-03

**Authors:** Merryn J. Netting, Najma A. Moumin, Emma J. Knight, Rebecca K. Golley, Maria Makrides, Tim J. Green

**Affiliations:** 1Women and Kids Theme, South Australian Health and Medical Research Institute, Adelaide, SA 5000, Australia; najma.moumin@sahmri.com (N.A.M.); emma.knight@sahmri.com (E.J.K.); maria.makrides@sahmri.com (M.M.); tim.green@sahmri.com (T.J.G.); 2Discipline of Paediatrics, Faculty of Health and Medical Sciences, University of Adelaide, Adelaide, SA 5000, Australia; 3Nutrition Department, Women’s and Children’s Health Network, Adelaide, SA 5006, Australia; 4School of Public Health, Faculty of Health and Medical Sciences, University of Adelaide, Adelaide, SA 5000, Australia; 5Caring Futures Institute, College of Nursing and Health Sciences, Flinders University, Adelaide, SA 5000, Australia; rebecca.golley@flinders.edu.au

**Keywords:** Australian feeding infants and toddler study, solid feeding, nutrient intakes, feeding practices, breastfeeding, allergen introduction

## Abstract

The Australian Feeding Infants and Toddler Study 2021 (OzFITS 2021) is a nationwide survey of Australian caregivers’ infant and toddler feeding practices. Here, we describe breastfeeding rates and duration, use of breastmilk substitutes, and introduction of complementary (solid) foods, including common food allergens. Caregivers (*n* = 1140) were recruited by a digital marketing company and were interviewed using a structured telephone questionnaire to obtain information. Breastfeeding was initiated in 98% of infants, but the duration of exclusive breastfeeding to six months was less than 1%. Nearly 40% of children continued to receive breastmilk beyond one year, with 10% of toddlers receiving breastmilk at two years. One-quarter of infants were introduced to solid foods between 4 to 5 months, and nearly all infants had received solid foods by 7 months. New guidelines encourage the early introduction of potential food allergens to reduce the risk of allergy, and by 12 months, over 90% of children had been given eggs and peanuts. One-third of children received no breastmilk substitutes during their first year. One-third of infants first received breastmilk substitutes following birth and before discharge from the hospital. Of these infants, 30% ceased breastmilk substitute use after discharge. Our findings suggest a high rate of continued breastfeeding with 44% receiving breastmilk beyond 1 year. One approach to increase the duration of exclusive breastfeeding is to reduce breastmilk substitute use while in hospital.

## 1. Introduction

The Australian National Health and Medical Research Council (NHMRC) Infant Feeding Guidelines recommend exclusive breastfeeding for the first six months to optimize infant growth, development, and health [1,2,3]. Afterward, infants should be given nutritious complementary (solid) foods with continued breastfeeding to 12 months and beyond. Solid foods should be introduced at around six months because breastmilk alone does not meet infant requirements for energy and select nutrients, particularly iron [3]. The type and order of solid foods introduced is not important as long as they are iron-rich and nutrient-dense [3], with increasingly complex textures matching the infant’s development.

Despite the importance of early infant feeding, there is very little data on breastfeeding practices and the timing of solid food introduction of Australian caregivers. The 2010 Australian National Infant Feeding Survey (ANIFS) [4] provides the most comprehensive assessment of early feeding against key breastfeeding indicators established by the World Health Organization and adapted for Australia. In the ANIFS, breastfeeding initiation was over 90%, with 27% of infants exclusively breastfed to four months. At 12 months, 40% of children were still receiving breastmilk. Nearly all infants had received solid foods by six months, with around 10% receiving them as early as three months. Because the ANIFS was conducted over a decade ago, breastfeeding patterns and the timing of solid food introduction will have likely changed. Furthermore, the ANIFS did not address the use and timing of the introduction of breastmilk substitutes. 

In their 2012 Infant Feeding Guidelines, the NHMRC [3] advised that foods or food allergens no longer needed to be avoided during infancy to prevent food allergies. In response to recent evidence [5], the Australasian Society of Clinical Immunology and Allergy (ASCIA) now encourages deliberate and regular inclusion of common allergens (e.g., egg and peanut) into infants’ diets before one year [6]. However, there are limited data on whether caregivers follow this advice. 

Here, we provide recent nationwide data on breastfeeding initiation, duration of exclusive, and the use of breastmilk substitutes among Australian caregivers. We also provide new data on the timing of introduction to solid foods, including early exposure to common food allergens. 

## 2. Materials and Methods

### 2.1. OzFITS 2021 Survey Methods

The methods for OzFITS 2021 are described in detail elsewhere in this supplement [7]. Briefly, caregivers with a child aged 0–24 months and who could answer questions about their dietary intake from birth were recruited using targeted social media advertisements. The sampling frame included all Australian states and territories, with the target sample proportional to population size. Telephone-based surveys and interviews were used to collect information. Participants (*n* = 1140) were surveyed between April 2020 and 2021. 

Questions regarding initiation of breastfeeding and feeds in the early neonatal period were consistent with the ANIFS [4]: “What was [CHILD] first feed?”, “How soon after [child] was born did he/she first have breastmilk?”, “While in hospital, did [child] ever have any other fluids or foods?”, “When [child] first came home from the hospital, was he/she receiving breastmilk?”. The timing of solid food introduction was determined by self-report. Caregivers were asked, “Has [CHILD] ever been given any food, including baby cereal added to bottles?” If they responded ‘yes’, they were asked what age solid foods were introduced. Introduction to the broad food groups and timing of common food allergen introduction (egg, peanut, tree nuts, and sesame) were also assessed using a questionnaire. 

Age definitions adopted were based on completed months, consistent with the ANIFS [4]. The definitions for breastfeeding based on WHO 2008 were adopted for use in the ANIFS survey (Table 1) [2,4]. 

### 2.2. Statistical Analysis

The OzFITS 2021 cross-sectional survey enrolled 1140 caregivers with children in four age bands: 0–5.9 months; 6–11.9 months; 12–17.9 months; and 18–23.9 months. A feature of the data when looking at events such as exclusive breastfeeding, predominant breastfeeding, and introduction to solid foods are censored responses: children who had not yet experienced the relevant event at the time of the survey. Kaplan–Meier survival analysis was used to estimate the cumulative proportion surviving at each event time and associated 95% confidence intervals. We estimated one minus the cumulative proportion of surviving at each event for solid food introduction. We were interested in the proportion who have experienced the event (e.g., started solid foods) rather than the proportion who have experienced a disqualifying event (e.g., stopped breastfeeding). Statistical analyses were completed using SPSS 28.0 (I.B.M. Corp., Armonk, NY, USA) [8].

## 3. Results

### 3.1. Breastfeeding

Exclusive breastfeeding was initiated in 93% of infants; however, only 59% were exclusively breastfed to one month, and 39% were exclusively breastfed to four months. Less than 1% of infants were exclusively breastfed to six months (Table 2). The main reason for exclusive breastfeeding cessation under one month of age was hospital exposure to breastmilk substitutes.

The median duration of any breastfeeding was 11.0 months (95% CI 10.2 to 11.8). Breastfeeding was initiated in 98% of infants, and at six months of age, 68% (95% CI 65 to 71) were breastfed. Nearly 40% of children were breastfed beyond 12 months, and at 20 months of age, 10% of toddlers were still receiving some breastmilk. 

### 3.2. Introduction to Breastmilk Substitutes

By six months of age, just over half of the infants had been given breastmilk substitutes, with 40% of infants introduced during the first month of life. By age one, 35% of children had never received breastmilk substitutes (Table 3). Of the 40% (*n* = 455) of infants consuming breastmilk substitutes in their first month of life, 78% (350/455) received these in hospital soon after birth. Of the infants who began breastmilk substitutes in the hospital, 33% (116/350) ceased consuming breastmilk substitutes after hospital discharge and received only breastmilk. Another 58% (202/350) continued to receive both breastmilk and a breastmilk substitute, and only 9% (32/350) received breastmilk substitutes alone. In contrast, of the children introduced to breastmilk substitutes at home by one month of age, 9% (9/105) ceased breastmilk substitutes and then only received breastmilk for a period; 81% (86/105) received both, and 10% (10/105) received only breastmilk substitutes. 

### 3.3. Timing of Introduction Complementary (Solid) Foods

The median age at first introduction to solid foods was 5 months (95% CI 4.9 to 5.1). One-quarter of all children were introduced to solid foods between 4 and 5 months, 33% commenced solids between 5 and 6 months, and by 6 to 7 months, 97% were introduced to solid foods. A small number (*n* = 13) started solid foods before 4 months. Of these, one child (<1 month of age) was given honey on their pacifier, one child (1–2 months) of age had pureed food at 4 weeks, and the remainder (*n* = 11) started solid foods at around 3 months and 3 weeks of age. Delayed food introduction beyond seven months was observed among 3% of infants (7–8 months *n* = 15; 8–12 months *n* = 15). Reasons for delaying the introduction to solid foods included prematurity *n* = 5; food intolerances or allergy *n* = 3; *n* = 22 parents gave no specific reason. 

### 3.4. Introduction to Common Allergens and Medically Diagnosed IgE-Mediated Food Allergy

By one year, around 95% of children had been exposed to the common food allergens hen’s egg and peanut, 76% of the children had consumed tree nuts, and 82% had consumed sesame (Table 4). The median age for introduction to egg was 6 months (95% CI 5.9 to 6.1), peanut was 7 months (95% CI 6.9 to 7.1), and sesame was 8 months (95% CI 7.7 to 8.3).

Amongst the children (*n* = 922) who had started solid foods, caregivers reported medically diagnosed IgE mediated food allergy, which was confirmed by allergen testing (skin prick test or allergen-specific IgE) in 77/922 children (8.5%). Of these, IgE-mediated egg allergy was reported in 2.4% (22/922), and peanut allergy was reported in 1.6% (15/922) of children.

## 4. Discussion

Here, we report the results of a recent, nationwide Australian survey of early infant feeding practices. We found that the duration of exclusive breastfeeding to 4 months was 40%, falling to less than 1% at 6 months. Our findings are similar to exclusive breastfeeding rates reported in the 2010 ANIFS [4], where exclusive breastfeeding to 4 and 6 months was 27% and 2%, respectively. Our findings also suggest that exclusive breastfeeding rates have remained unchanged during the last decade. 

Our exclusive breastfeeding duration differs from that reported in the Australian Bureau of Statistics, 2017–2018 National Health Survey of Breastfeeding, where 61% and 29% of children were exclusively breastfed to 4 and 6 months, respectively [9]. This difference is likely due to differences in the survey questions used in the two studies. In the Australian Bureau of Statistics Survey, a single question was asked to determine the rate of exclusive breastfeeding. In the ANIFS and our survey, caregivers were probed for breastmilk substitute exposure while in hospital and after hospital discharge, which may have led to an overestimation of exclusive breastfeeding rates in the Australian Bureau of Statistics. Survey. 

Of concern is that one-third of infants in our study lost their exclusive breastfeeding status due to exposure to breastmilk substitutes while in hospital. Of these infants, 30% stopped consuming breastmilk substitutes after discharge and received only breastmilk for a time. These findings are consistent with a Victorian (Australia) state-wide perinatal indicators report where 29% of infants were given breastmilk substitutes in hospital [10]. Further studies are required to determine the reasons for early introduction to breastmilk substitutes in hospitals. Strengthened antenatal education and increased support for breastfeeding in the early neonatal period may help decrease breastmilk substitute use in hospitals [1]. The Baby-Friendly Hospital Initiative [11] was established to increase exclusive breastfeeding rates, but only 26% of Australian hospitals are accredited [12].

Despite the short duration of exclusive breastfeeding in OzFITS 2021, breastfeeding duration to one year was high, with many children breastfed into their second year of life. Our finding of long breastfeeding duration is consistent with other Australian cohorts and surveys [13,14]. In the 2017–2018 National Health Survey of Breastfeeding, 40% of children were breastfed to at least 12 months of age compared to 44% of OzFITS 2021 participants. Although half of the infants in our survey had consumed breastmilk substitutes, one-third of the infants surveyed had never received breastmilk substitutes. Among those who had received breastmilk substitutes, 86% continued to receive some breastmilk, suggesting that many caregivers use them as an addition to, rather than a replacement of breastmilk. There are no comparable Australian data on the use of breastmilk substitutes.

The Australian NHMRC infant feeding guidelines [3] recommend introducing solid foods at around 6 months when the baby is developmentally ready, and by 6 months of age, 96% of our cohort had commenced solid foods. About 25% of infants in OzFITS 2021 were introduced to solid foods at 4–5 months, increasing to two-thirds by 5–6 months. Due to the higher risk of developing an allergy, the Australasian Society of Clinical Immunology and Allergy (ASCIA) advises that solid foods are not introduced before 4 months [5,6]. We found that very few caregivers (1.5%) had introduced solid foods before 4 months of age, 25% (95% CI 23 to 28) introduced at 4 months, and most children (58.1%; 95% CI 55 to 61) had commenced on solid foods between 5 and 6 months of age. The 2010 ANIFS results indicate that 35% of infants surveyed had received solid food at 4–5 months [4]. Results from several other Australian surveys suggest that the timing of solid foods seems to be moving more to the 5 to 6th month, with fewer starting solid foods very early (<4 months of age) [14,15,16].

In OzFITS 2021, infants were introduced to common allergens such as egg and peanut by one year of age, consistent with results reported elsewhere [17,18,19]. We also examined tree nuts and sesame timing in babies’ diets, as these are common food allergens in Australian children [20]. Furthermore, the hospital exposure to breastmilk substitutes and subsequent cessation may increase the risk of cow’s milk allergy [21]. The prevalence of IgE-mediated allergy reported amongst our cohort was consistent with that reported [20].

A strength of this study is that it provides contemporary Australian data with a sample proportionate to the population size within each state. Questions were adapted from the ANIFS but included more comprehensive questions regarding the timing of solid foods and common food allergens. Limitations include the retrospective nature of the feeding survey data collected, specifically for the older age group. We used convenience sampling, which is more subject to response bias than purposeful sampling. In our case, participants were more highly educated and had a higher family income than the Australian population [7,22]. Despite this, our early feeding data are consistent with the ANIFS and other Australian feeding studies reported. 

In conclusion, for OzFITS 2021, we report exclusive breastfeeding rates at 4 and 6 months compared with the ANIFS 2010. Most caregivers commenced solid foods at around 6 months of age, according to the NHMRC Infant Feeding Guidelines, and most infants were exposed to common allergens in the first year of life. Perhaps the most worrying finding is the early introduction to breastmilk substitutes in hospitals, affecting exclusive breastfeeding rates. Many caregivers surveyed breastfeeding, ceased breast milk substitutes, and gave their infant only breastmilk, possibly suggesting they intended to exclusively breastfeed their babies Strategies that support exclusive breastfeeding while in hospital are needed. 

## Figures and Tables

**Table 1 nutrients-14-00206-t001:** Definitions of infant feeding practices used for OzFITS 2021 ^1^.

Feeding Practice	Requires that the Infant Receive	Allows the Infant to Receive	Does Not Allow the Infant to Receive
Exclusive breastfeeding	Breastmilk (including expressed milk)	Oral rehydration solutions, drops, syrups (vitamins, minerals, medicines)	Anything else
Predominant or ‘full’ breastfeeding	Breastmilk (including expressed milk) as the predominant source of nourishment	Certain liquids (water and water-based drinks, fruit juice), and oral rehydration salts, drops, or syrups (vitamins, minerals, medicines)	Anything else
Solid feeding or ‘partial’ breastfeeding	Breastmilk (including expressed milk)	Anything else	
Any breastfeeding	Any of the above definitions		
Ever breastfed	Breastfed or received expressed breastmilk at least once	Anything else	

^1^ Adapted from the Australian National Infant Feeding Survey [4].

**Table 2 nutrients-14-00206-t002:** Cumulative proportion of children exclusively breastfed, predominately breastfed, or receiving any breastmilk by month of age.

Age (to Month)	Equivalent Duration	Exclusively Breastfed ^1^	Predominantly Breastfed ^2^	Receiving Any Breastmilk
		% (95% CI)		
0 to <1	Less than 1 month	59 (56, 62)	60 (57, 63)	98 (97, 99)
1	Less than 2 months	57 (54, 60)	58 (55, 60)	95 (94, 96)
2	Less than 3 months	54 (51, 57)	55 (52, 58)	91 (90, 93)
3	Less than 4 months	51 (48, 54)	52 (50, 55)	87 (85, 89)
4	Less than 5 months	39 (36, 42)	40 (37, 43)	82 (79, 84)
5	Less than 6 months	22 (19, 24)	23 (20, 26)	75 (72, 77)
6	Less than 7 months	1 (0, 1)	1 (0, 2)	68 (65, 71)
7	Less than 8 months	0 (0, 1)	0 (0, 1)	63 (60, 66)
8	Less than 9 months	0 (0, 0)	0 (0, 1)	60 (57, 63)
9	etc.	-	-	54 (51, 57)
10		-	-	50 (46, 53)
11		-	-	47 (44, 51)
12		-	-	44 (40, 47)
13		-	-	39 (35, 42)
14		-	-	34 (30, 37)
15		-	-	30 (27, 33)
16		-	-	27 (24, 30)
17		-	-	22 (19, 25)
18		-	-	17 (14, 20)
19		-	-	14 (11, 17)
20		-	-	10 (8, 13)
21		-	-	6 (4, 9)
22		-	-	4 (3, 6)
>23		-	-	4 (3, 6)

^1^ Allows breastmilk, expressed milk, oral rehydration solutions, vitamins, and minerals. ^2^ Allows breast milk as the only milk source; breast milk substitutes or other kinds of milk are not permitted. Oral rehydration solutions, vitamins and minerals, water, and juice are permitted.

**Table 3 nutrients-14-00206-t003:** Use of breastmilk substitutes and first exposure to solids foods reported as a cumulative proportion by age in months.

Age (Months)	Introduced to Breastmilk Substitute	Introduced to Solid Foods
	% (95% CI)	
<1 month	40 (37, 43)	0 (0, 1)
1 to <2 months	42 (40, 45)	0 (0, 1)
2	45 (42, 48)	0 (0, 1)
3	48 (45, 51)	1 (1, 2)
4	51 (48, 54)	25 (23, 28)
5	53 (50, 56)	58 (55, 61)
6	56 (53, 60)	97 (95, 98)
7	57 (54, 60)	98 (97, 99)
8	60 (57, 63)	99 (99, 100)
9	61 (58, 65)	99 (99, 100)
10	64 (61, 67)	100 (99, 100)
11	65 (62, 68)	100 (99, 100)
12	66 (63, 70)	100 (99, 100)
>12	67 (64, 71)	100 (99, 100)

**Table 4 nutrients-14-00206-t004:** First exposure to common food allergens reported as a cumulative proportion by age in months.

Month	Egg	Peanut	Tree Nuts	Sesame
	% (95% CI)			
Birth to <1	0 (0, 0)	0 (0, 1)	0 (0, 0)	0 (0, 0)
1	0 (0, 0)	0 (0, 1)	0 (0, 0)	0 (0, 0)
2	0 (0, 0)	0 (0, 1)	0 (0, 1)	0 (0, 0)
3	0 (0, 1)	0 (0, 1)	0 (0, 1)	0 (0, 0)
4	3 (2, 4)	3 (2, 5)	1 (1, 2)	1 (0, 2)
5	12 (10, 14)	13 (11, 15)	6 (4, 7)	5 (4, 6)
6	54 (50, 57)	50 (47, 53)	22 (19, 25)	22 (19, 25)
7	71 (68, 74)	67 (64, 70)	37 (33, 40)	37 (34, 40)
8	82 (79, 85)	78 (75, 81)	49 (46, 53)	51 (47, 54)
9	87 (85, 90)	83 (80, 86)	56 (52, 59)	59 (55, 62)
10	92 (90, 94)	87 (85, 90)	62 (59, 66)	66 (63, 70)
11	93 (91, 95)	88 (86, 90)	65 (61, 68)	69 (65, 72)
12	97 (95, 98)	94 (93, 96)	76 (72, 79)	82 (79, 85)
>12	98 (97, 99)	98 (97, 99)	87 (84, 90)	91 (89, 93)

## Data Availability

The data presented in this study are available on request from the corresponding author.

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
