# Peer review of "The Australian Feeding Infants and Toddler Study (OzFITS 2021): Breastfeeding and Early Feeding Practices"

_nutrients, 2022, doi:10.3390/nu14010206_

Round 1
Reviewer 1 Report
Thank you very much for giving me the opportunity to review this work. Any support to promote breastfeeding is relevant. However I have some suggestions to improve the manuscript:
The abstract can be improved by adding the methodological aspects of the study.
The Introduction should be completed and improved. There are claims without sufficient theoretical foundation.
The methodology is not sufficiently explained. Although it says that it appears in another work, it is not known if it is published, as it is not referenced.
The statement if infants had not received substitute breast milk, the duration of exclusive breastfeeding would be longer should be clarified by providing appropriate appointments.
Line 167 correct typographical error. Lines 168 and 169 there is a repeated phrase.
Line 192, please define ASCIA
Although it states in the discussion that the sample size is proportional to each state, it is not clear in the methodology how it has been established.
Reviewer 2 Report
This is a well-written manuscript that flows well from methods to results. It is an enjoyable and stimulating read.
Line 52 This section is about why your study is needed. You wrote, "have likely changed." Consider rewriting to say an update is appropriate.
Line 52- To make your point about the need for current data on breastfeeding and introducing solids, consider describing the most recent NHMRC Infant Feeding Guidelines in reference to introducing solid foods about exposure to common allergens. Spell out NHMRC Health and Medical Research Council (NHMRC), Canberra. Thereafter, use NHMRC. (Introduction, Line 216, and Line 188).
Lies 164-170 This is an important section for this manuscript because it is your contention that breastmilk substitutes in hospital are problematic (as you report in Lines 217-221). The sentence structure issues make it hard to follow. Please review and rewrite. Explain specifically how your data directly support your conclusion that early introduction in hospitals affects exclusive breastfeeding rates.
Line 188 Specifically name NHMRC Infant Feeding Guidelines.
Lines 201-202 Please include a citation for why you included sesame in your research objectives.
Lines 210-211 Explain why your sample was biased toward higher income groups and suggest what might be done in future breastfeeding studies to reduce this bias. Cite the source, "As typical for many studies, there was a bias towards higher-income groups."
Lines 217-221 Because you address NHMRC Infant Feeding Guidelines specifically in conclusions, consider specifically addressing their guidelines about solid foods in the introduction.
